# Multi-Source Satellite and WRF-Chem Analyses of Atmospheric Pollution from Fires in Peninsular Southeast Asia

**Ailin Liang** ⬤, **Jingyuan Gu** * and **Chengzhi Xiang**

School of Remote Sensing & Geomatics Engineering, Nanjing University of Information Science & Technology, Nanjing 210044, China; ireneliang@nuist.edu.cn (A.L.); xcz0726@nuist.edu.cn (C.X.)
* Correspondence: 20211248046@nuist.edu.cn

**Abstract:** Atmospheric pollutant gases emitted from straw burning and forest fires can lead to air quality and human health problems. This work explored the evolutionary trends of atmospheric $CO_2$ and other pollutant gases in five countries of Peninsular Southeast Asia (PSEA) over a long time series using various satellite remote sensing data. The research results indicate that a considerable number of fires occur in the region every spring, which negatively affects air quality. The concentration of $CO_2$ increased every year, indicating a correlation coefficient of 0.57 with the number of fire points. The concentration of CO and $NO_2$, respectively, showed a correlation coefficient of 0.87 and 0.95 with the number of fire points as well. Additionally, the AOD reflects the relationship between fire points and air quality. The study also used the meteorological and air quality Weather Research and Forecasting with Chemistry (WRF-Chem) to simulate the fire season in March 2016. In this sensitivity study, we examined the impact of air pollutant gases on air quality in PSEA under a hypothetical scenario with and without fire emissions. The simulation results were also compared with satellite observations, which showed that the WRF-Chem model and the FINN (Fire INventory from NCAR) inventory could effectively simulate the spatial distribution and spatial–temporal variability characteristics of CO concentration in the fire, but the simulation result of $NO_2$ was not satisfactory. This study suggests that spring wildfires affect not only air quality, but also short-term weather in the region.

**Keywords:** air pollutant gases; biomass burning; Peninsular Southeast Asia; OCO-2; Sentinel-5P; WRF-Chem

## 1. Introduction

Biomass burning, mainly caused by forest fires and straw burning, has a significant impact on global air quality, cloud cover, atmospheric radiation, the water vapour cycle and several aspects of regional climate [1]. South and Southeast Asia is one of the world's top three biomass burning sites. Every spring, wildfires in PSEA (PSEA: Cambodia, Laos, Myanmar, Thailand, Vietnam; PSEA is also referred as Indochina in some cultures) are exceptionally intense [2,3]. The entirety of Southeast Asia belongs to the tropical and subtropical climate zones, with most of the annual precipitation being considerable. It is subject to a wide and regular monsoon weather system, producing significant wet seasons (from June to October) and dry seasons (from November to May) in most parts of the region. The analysis in terms of the number of fire points, discriminating also for each country (Figure 1) and the fire frequency map in PSEA over the years (Figure 2), can be obtained by using the product MCD14DL from 2015–2021 MODIS data. The most affected country was Myanmar, where 67,076 fire events occurred from 2015 to 2021, followed by Laos (47,005), Cambodia (43,067), Thailand (20,014) and Vietnam (9327). Since 2018, there has been a significant increase in the number of fire points, about 61% more in 2019 (51,458) than in 2018 (31,857), which is partly due to human-induced fires set by local people [4], and partly caused by fires in forest grassland areas [5]. To gain a deeper comprehension of the atmospheric burden incurred by these fires (both in the present and future), we

integrate satellite monitoring techniques with chemical modelling to fully examine the added pollution from wildfires in PSEA.

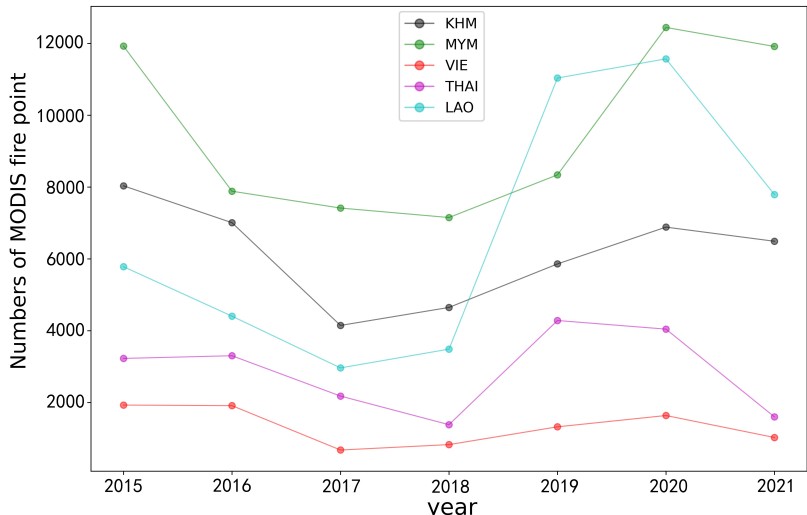

**Figure 1.** Statistics on the number of fire points in Peninsular Southeast Asian countries, 2015–2021.

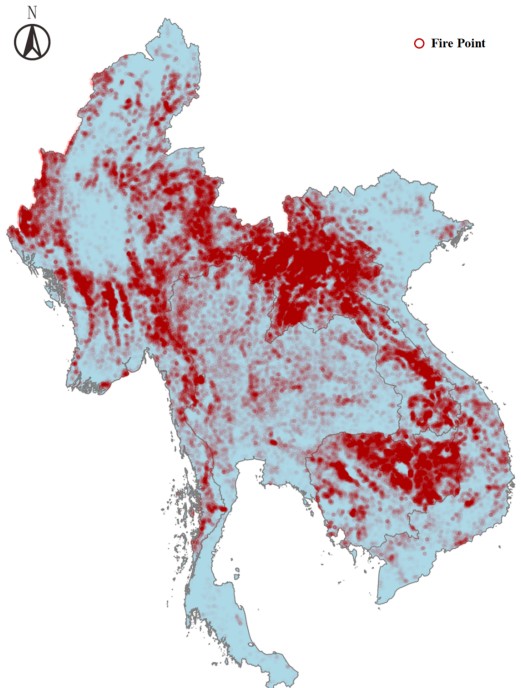

**Figure 2.** Fire frequency in Southeast Asian countries, 2015–2021.

The Weather Research and Forecasting model coupled with Chemistry (WRF-Chem) has been widely used for the investigation of air quality on a regional scale [6]. The model works in online mode and can simulate different atmospheric pollutants [7,8], both natural and anthropogenic. Most of the previous studies have focused on the atmospheric chemistry and physics of air pollutants [9,10] and meteorological conditions [11]. For example, Sharma et al., 2022 [12] and Nguyen et al., 2021 [13] simulated the scenarios with and without wildfire. Rizza et al., 2021 [14] used WRF-Chem to investigate the effects of variable eruption source parameters on volcanic plume transport in the Mediterranean basin after the paroxysm of Mount Etna. Grell et al., 2011 [11] found that the interaction between aerosols and atmospheric radiation following the fire season caused considerable

changes in temperature and humidity. Only a few methods have been used to assess and quantify the impact of biomass burning on air pollution in PSEA. For example, sensitivity tests conducted by Lee et al. [15] with WRF-Chem showed that in PSEA, sulphate emissions could be reduced by 25% if natural gas was used to replace coal in the power generation and industrial sectors, and black carbon concentrations could be reduced by 42% if natural gas was used to replace biofuels in the residential sector. Dong and Fu 2015 [3] investigated the interannual variations of biomass burning from PSEA in terms of its emission, transport and impacts over the local and downwind areas from 2006 to 2010.

The modelling analysis method can only quantitatively assess the pollution contribution of a fire, while the comprehensive assessment of its impact on air quality requires further incorporation of multi-source satellite data over long time scales. Therefore, this study used multi-source remote sensing satellite data to analyse the changes of fire points, $CO_2$, CO, $NO_2$ and aerosols during forest fires in PSEA, with the aim of seeking the relationship between trace gases and fire points, as well as quantifying the impact of fire smoke on air quality. Furthermore, the WRF-Chem model, Version 3.9.1, was used to simulate the atmospheric particulate mass concentrations and their distribution in PSEA without and with fire conditions. The results under the two conditions were then compared to discern the impact of fire on air quality, and further validated with satellite data.

## 2. Materials and Methods

In this section, we will first describe the various types of data used, and then introduce some aspects of WRF-Chem, in particular the coupling of chemistry, radiation, and cloud microphysics. Next, we will detail the experimental setup.

### 2.1. Data

#### 2.1.1. Observational Data

The observational data used in this work include meteorological observations and satellite observation data. Meteorological observations were derived from (https://rp5.ru/, accessed on 10 October 2023), including surface air temperature, relative humidity and barometric pressure, and the stations selected for this paper were surface meteorological observation stations in six cities, including Bangkok, Chiang Mai, Vientiane, Phnom Penh, Ho Chi Minh and Mandalay. The position of the observations in PSEA is shown in Figure 3.

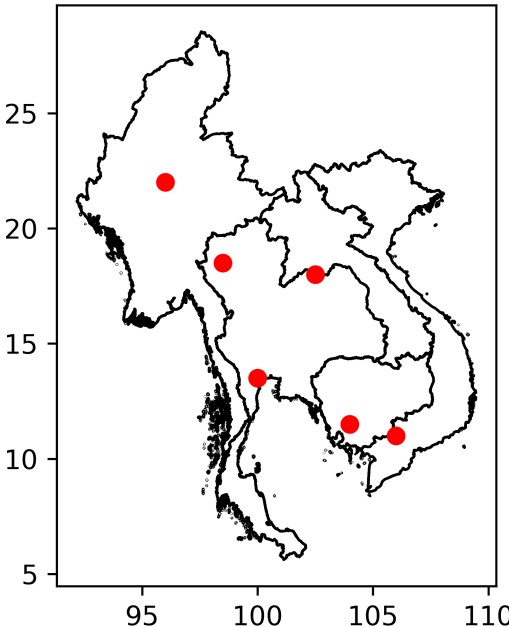

**Figure 3.** Meteorological observations in PSEA.

2.1.2. Satellite Data

The Moderate Resolution Imaging Spectroradiometer (MODIS) satellite fire point and aerosol optical depth (AOD) products, OCO-2 satellite, Sentinel-5P and MOPITT satellite data were used in this work.

MODIS NRT C6.1 (MCD14DL) is NRT MODIS (Terra and Aqua) Collection 6.1 data processed by NASA LANCE FIRMS. The thermal anomaly/fire location represents the centre of a 1 km pixel, marked by the MODIS MOD14/MYD14 fire and thermal anomaly algorithm to contain one or more fires within the pixel. This is the most basic fire product for identifying active fires and other thermal anomalies such as volcanoes. Each thermal anomaly/fire location point included multiple fields of information such as latitude, longitude, brightness, acquisition time, confidence, fire radiated power, etc. The MCD14DL product was provided by LANCE FIRMS (https://earthdata.nasa.gov/earth-observation-data/near-real-time/firms/c6-mcd14dl, accessed on 10 October 2023) for distribution and was available in txt, Shapefile (SHP) and Keyhole markup language (KML) formats. We screened points with a confidence of greater than 80% from 2015 to 2021 and designated them as fire points to provide data for this study.

The OCO-2 satellite was launched and commissioned in July 2014 by NASA. It is capable of providing high-spatial-resolution ground-based $XCO_2$ observations [16]. Its purpose is to accurately observe carbon emissions and cycles, improve global carbon cycle observation models and better understand changes in atmospheric $XCO_2$ in order to more accurately predict global climate change trends [17]. Currently, the daily observed L2 $XCO_2$ data from OCO-2 satellite released on the official website had a spatial resolution of 2.25 km × 1.29 km and a revisit period of 16 days. The L2 $XCO_2$ data from 2015 to 2021 in the PSEA were selected (The data of August 2017 were missing, and those of July and September 2017 were selected for mean interpolation) to analyse the concentration of $CO_2$ and its underlying changes in the research area.

MCD19A2 is MODIS series aerosol product data with a spatial resolution of 1 km and a temporal resolution of 1 day, providing wide coverage and high inversion accuracy, available at (https://code.earthengine.google.com/?scriptPath=Examples:Datasets/MODIS/MODIS_061_MCD19A2_GRANULES, accessed on 10 October 2023). It is generated with the aerosol algorithm in MAIAC (Multi-Angle Atmospheric Correction Algorithm) and is currently the latest C6 version of the product. It has advantages over the Deep Blue and Dark Target algorithms, providing global atmospheric data to monitor aerosol characteristics and dynamics. In this paper, the monthly values of AOD in the 470 nm band from the MCD19A2.006 dataset on the GEE platform for the years 2015–2021 were selected in this paper.

The MOPITT sensor was carried on board the NASA Terra satellite and lifted off in December 1999 to operate at an altitude of 705 km above the earth with a repeat observation period of 3 days and a horizontal resolution of 22 km. In this paper, the monthly mean data of the MOPITT TIR+NIR CO inversion values from March versus September 2016 (MOP03JML3V9) were used as a validation of the WRF-Chem simulation experiment for comparison. The data were available for download at (https://asdc.larc.nasa.gov/data/MOPITT/MOP03JM.008, accessed on 10 October 2023) in he5 format with a spatial resolution of 1° × 1°. The latitude, longitude, time and vertical column concentration of tropospheric CO in PSEA were extracted by Python programming.

The Sentinel-5P satellite features the highly advanced TROPOMI sensor, which has the capability to measure ultraviolet and visible light in the range of 270 to 500 nm, near-infrared in the range of 675 to 775 nm and shortwave infrared in the range of 2305 to 2385 nm. The TROPOMI subsatellite with a spatial resolution of an unprecedented 3.5 × 7 km enables more accurate imaging of a wide range of air pollutants such as $NO_2$, $O_3$, $SO_2$, $CH_4$ and CO. The CO and $NO_2$ data products selected in this paper were level 3 datasets of Sentinel-5P OFFL on the GEE platform from 2018 to 2021, available at (https://developers.google.com/earth-engine/datasets/catalog/COPERNICUS_S5P_NRTI_L3_CO, accessed on 10 October 2023), respectively, which provide offline high-resolution imagery of $NO_2$ and CO.

### 2.2. Model Description and Configuration

In this study, two simulations were carried out by using the WRF-Chem Version 3.9.1 (available at the following link: https://www2.mmm.ucar.edu/wrf/users/download/get_source.html, accessed on 10 October 2023). The adopted domain (see Figure 4), covering five Southeast Asian countries and part of China, was composed of $80 \times 100$ grid points, with a horizontal resolution equal to 27 km. The vertical domain was composed of 38 layers from the surface up to 50 hPa.

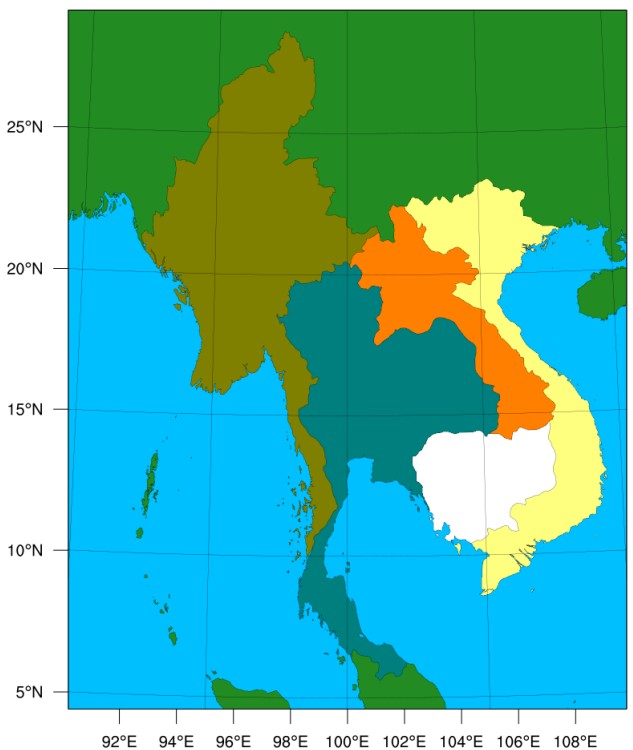

**Figure 4.** WRF-Chem simulation area settings.

The initial and boundary conditions of the meteorology in the model were driven by the NCEP FNL (Final Operational Global Analysis) data (available at the link: https://rda.ucar.edu/datasets/ds083.2/, accessed on 10 October 2023) with a temporal resolution of 6 h and a spatial resolution of 1 degree. The boundary conditions for air quality were provided by Atmosphere Model with Chemistry (CAM-Chem) [18] (available at the link: https://www.acom.ucar.edu/cam-chem/cam-chem.shtml, accessed on 10 October 2023) with a practical resolution of 6 h and a range of variables covering meteorological elements and atmospheric material concentrations. The choice of parameterization scheme in the model simulation is shown in Table 1, together with Grell-3 [19] for cumulus parameterization, Morrison 2-moment [20] for the microphysics scheme, RRTMG [21] for the longwave and shortwave radiation scheme, MM5 Monin Obukhov [22] for the surface layer scheme, MYNN 2.5level TKE [23] for the Planetary boundary layer scheme and Fast-J photolysis [24] for the photochemical scheme. The gas-phase chemical mechanism in the model simulation of the present study SAPRC99 [25] contained 74 reactants and 211 reaction equations. The emission inventories used in the model simulation were the Model of Emissions of Gases and Aerosols from Nature (MEGANV2.1) [26] and the biomass combustion emission inventory from NCAR's Fire Inventory from NCAR(FINNV1.5) [27].

**Table 1.** WRF-Chem configuration.

| Schemes | Parameterization Options |
|---|---|
| Microphysics | Morrison 2-moment |
| Long-wave radiation | RRTMG |
| Short-wave radiation | RRTMG |
| Cumulus parameterization | Grell-3 |
| Boundary layer scheme | MYNN 2.5level TKE |
| Surface layer | MM5 Monin-Obukhov |
| Photochemical | Fast-J photolysis |
| Gas-phase chemical mechanisms | SAPRC99 |
| Aerosol mechanism | MOSAIC |
| chem_opt | =203 |

Two one-month simulations, a base run including fire emissions and another run without fire emissions, were conducted for March 2016 using WRF-Chem. Both simulations were run from 1 March 2016 to 30 March 2016 with a time step of 120 s, providing a spin-up time of 48 h. The model outputs were generated and stored every 1 h for analysis to assess the sensitivity of air quality forecasts to emissions.

## 3. Results

### 3.1. Analysis of Satellite Observations

3.1.1. Changes in $CO_2$ during the Fire

The amount of $CO_2$ released from biomass combustion is considerable and will inevitably affect air quality. The spatial distribution of seasonal means of $XCO_2$ for the months of January, April, July and November 2016 were selected, as displayed in Figure 5. The mean value of $CO_2$ in January, April, July and November was 404.54 ppm, 405.27 ppm, 402.81 ppm and 402.14 ppm, respectively.

In other years, there were also higher concentrations of $CO_2$ during the winter and spring months than those during the summer and autumn months. Some fires also occurred in parts of the research area each year after the dry season in June. For instance, from July to October in 2015 there were 707 fires in the research area, and from July to October in 2016 there were 313 fires, but this was much lower than the number of fires during the fire season. Therefore, in the absence of fires, the average $CO_2$ concentrations were at a low level. Intensive fires broke out at the beginning of November, and the $CO_2$ concentrations rose again more slowly. By mid-December, as the number of fires increased significantly, the regional $CO_2$ concentration also increased significantly, and in the following spring, as plant respiration was enhanced, the $CO_2$ concentration also increased, and the number of fires in the region reached its maximum. The simultaneous effects of plant respiration and biomass combustion increased the concentration of $CO_2$ when there was a fire by about 6 ppm compared to the time when there was no fire (Figure 6). It can be found from the figure that the $CO_2$ concentration curve (blue curve) exhibited a slight lag compared with the curve of the number of fires (red curve), and the extreme values of the number of fires and $CO_2$ concentration did not emerge in the same time period primarily because of the respiratory and photosynthetic effects of terrestrial plants and animals, with the influence of fires being secondary.

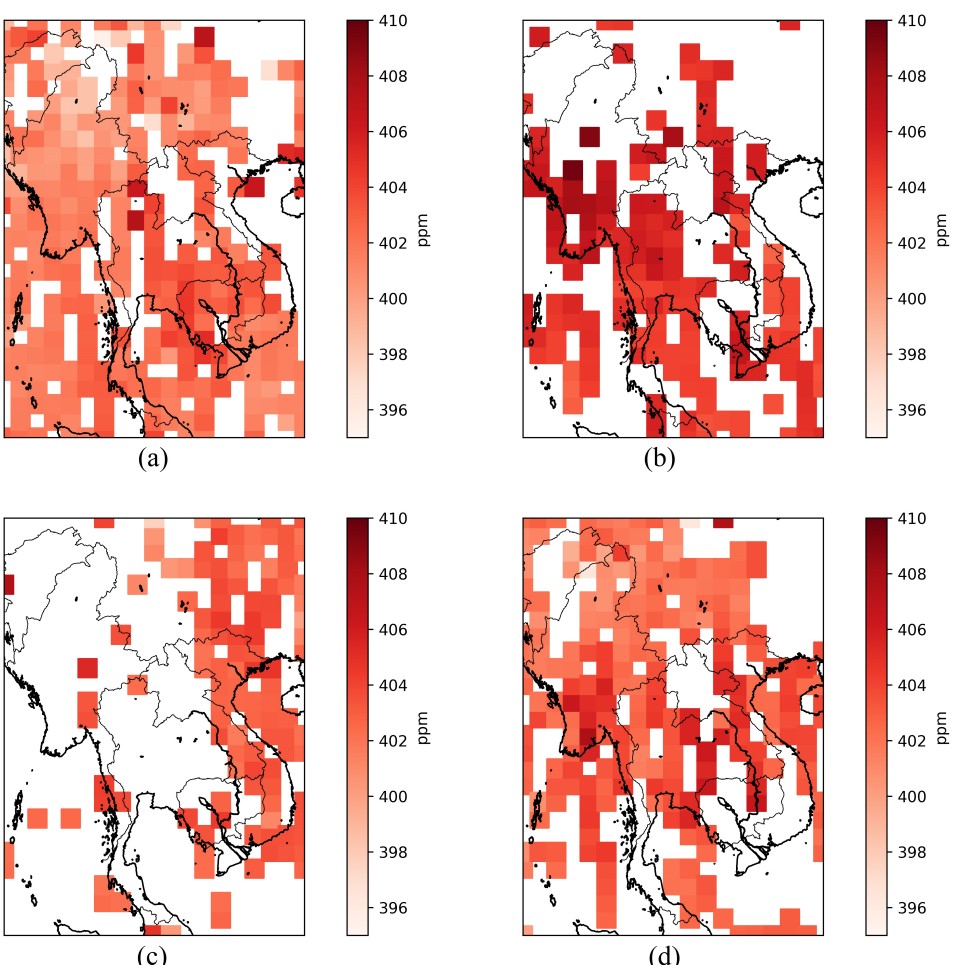

**Figure 5.** Spatial distribution of monthly mean values of atmospheric $XCO_2$ in the study area in 2016: (**a**) January; (**b**) April; (**c**) July; (**d**) November.

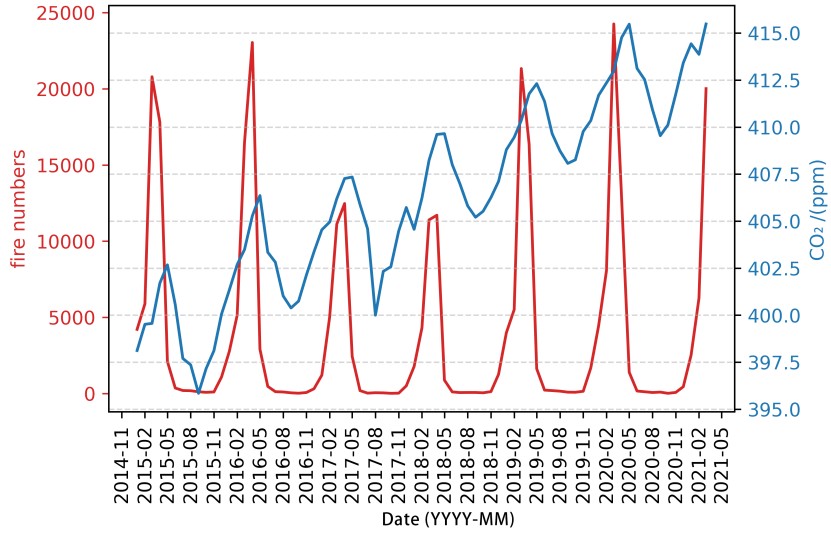

**Figure 6.** Plot of number of fires and $CO_2$ concentration over time.

In order to further study the relationship between the monthly average concentration of $CO_2$ and the number of fire points, we carried out a correlation analysis on the difference between the fire points and the peak–trough values of $CO_2$ for each year. Table 2 compares

the changes in the number of fire points and the fluctuation of $CO_2$ for each year. As can be seen from the table, the Pearson's correlation coefficient R between the two was 0.57, indicating their weak correlation.

**Table 2.** Correlation between the number of fires and the extreme difference in interannual $CO_2$ concentrations.

| Vintages | $CO_2$ Concentration Change (Unit: ppm) | Number of Fires (Unit: one) |
|---|---|---|
| 2015 | 6.83 | 52,984 |
| 2016 | 5.98 | 51,540 |
| 2017 | 5.01 | 33,189 |
| 2018 | 4.41 | 31,857 |
| 2019 | 4.239 | 51,548 |
| 2020 | 5.93 | 51,975 |
| 2021 | 5.215 | 41,638 |

The seasonal and trend decomposition using loess (STL) is a very general and robust method for decomposing time series. The STL time series decomposition method uses locally weighted regression as a smoothing method to decompose the time series into trend terms, seasonal terms and irregular residual terms [28]:

$$Yv = Tv + Sv + Rv \tag{1}$$

where Yv is the observed value at the moment of V; Tv, Sv and Rv are the trend term, seasonal term and residual term at the moment of V, respectively. The STL time series decomposition can reveal the trend, cycle length and random fluctuation range of $CO_2$ with the time series. As shown in Figure 7, due to the long lifetime and easy accumulation of $CO_2$, $CO_2$ generally gradually increased from 2015 onwards, with an obvious periodicity (the period was 12 months), reaching a great value in spring and a very small value in autumn, and it can be seen from the residual term that the residual term in the front of the time series fluctuated slightly more significantly than that in the back of the time series, which suggests that $CO_2$ was weakened by the disturbances caused by fire elements.

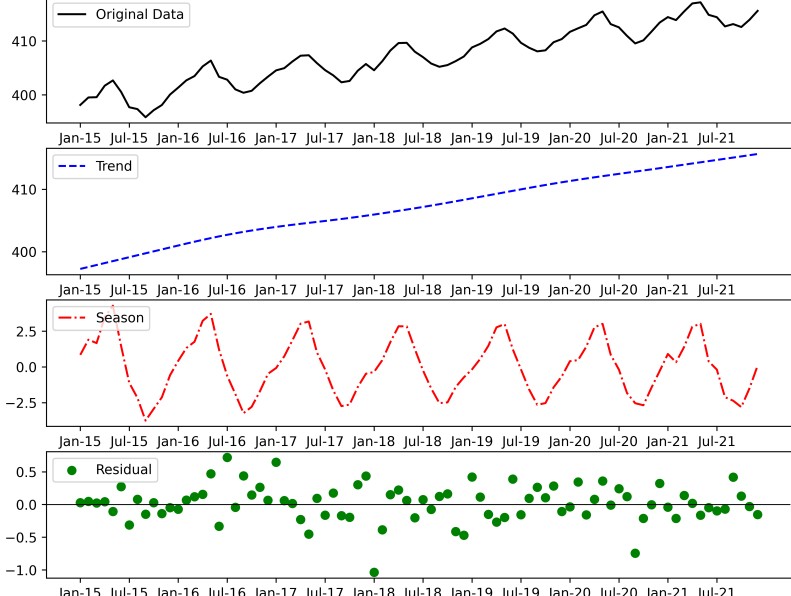

**Figure 7.** Schematic representation of STL decomposition results of $CO_2$ time series, 2015–2021.

### 3.1.2. Changes in CO during the Fire

Unlike $CO_2$, which has a long residence time in the atmosphere, CO, with a chemical lifetime of weeks to months, is often used as a conservative tracer to reflect the state of fires, with about 36% of global surface CO emissions coming from fires [29]. We used the Sentinel-5P high-precision atmospheric environmental monitoring satellite to comprehensively evaluate the atmospheric CO pollution from fires in Southeast Asia since 2018. In the fire-free month of July 2018, the monthly average CO concentration in the research area was $1.63 \times 10^{18}$ (molecule $\times$ cm$^{-2}$). In the early stage of the fire season, in December 2018, the CO concentration in the eastern coastal region fluctuated around $2.11 \times 10^{18}$ (molecule $\times$ cm$^{-2}$), and with the emergence of sporadic fires (1261 fires), there was a slow increase in CO concentration from January 2019 onwards, with the monthly mean of CO concentration increasing from $2.41 \times 10^{18}$ (molecule $\times$ cm$^{-2}$) in January to $3.13 \times 10^{18}$ (molecule $\times$ cm$^{-2}$) in March and the number of fires increased from 3997 in January to 21329 in March. However, the fires weakened into April 2019, and the number of fire points began to decrease incrementally, accompanied by a reduction in the CO concentration, until the fire season passed completely in June, when the CO concentration returned to the normal level of $1.81 \times 10^{18}$ (molecule $\times$ cm$^{-2}$), and the number of fire points also decreased to 230, which suggests that forest fire straw burning was responsible for the significant increase in CO concentration.

As shown in Figure 8, the time series plot of CO concentration (blue curve) changed with the number of fire points (red curve), the extreme value of CO concentration and the number of fire points always emerged at the same time and the change of CO concentration exhibited a U-shaped trend, which indicates that CO could serve as an effective indicator to evaluate the pollution degree of the atmosphere caused by local fires.

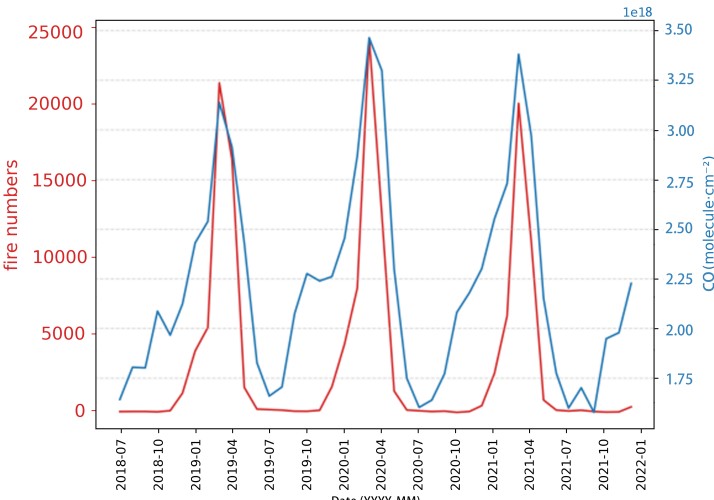

**Figure 8.** Plot of the number of fire points and CO concentration over time.

To further investigate the relationship between the monthly mean CO concentration and the number of fire points, we performed a correlation analysis between the number of fire points every month and the difference between the monthly CO concentration and the annual mean. Figure 9 compares the correlation between the number of fire points per month and the CO distance from the mean. The two were strongly correlated, with a Pearson correlation coefficient R of 0.87.

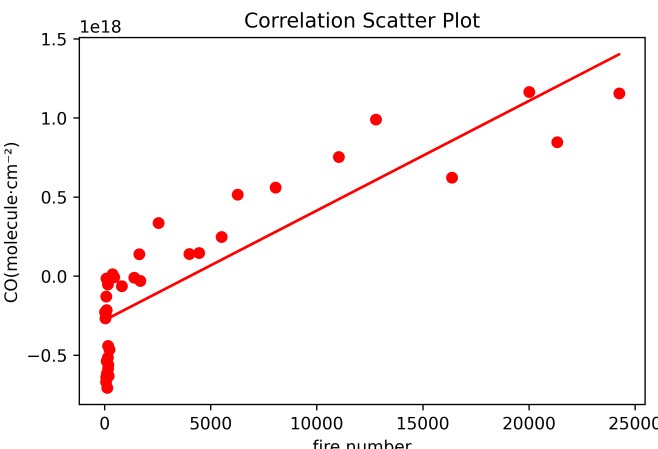

**Figure 9.** Correlation between the number of fires per month and the CO distance level.

### 3.1.3. Changes in NO$_2$ during the Fire

The EMI satellite payload monitoring tropospheric NO$_2$ concentrations during the 2019 Australian fires showed a strong correlation between NO$_2$ concentrations and the number of fire spots and degree of combustion during the fires as reported in a previous study [30]. Ilann Bourgeois et al. [31] measured hundreds of plumes from summer wildfires in the United States on flights in a DC-8 aircraft, and found that the fire smoke contained a significant amount of NO$_2$. We used the high-resolution Sentinel- 5P satellite Level 3 offline data product to calculate monthly averages of NO$_2$ concentrations in the study area. The results showed that the average values in 2019, 2020 and 2021 were $1.15 \times 10^{15}$ (molecule $\times$ cm$^{-2}$); the 2020 average was $1.078 \times 10^{15}$ (molecule $\times$ cm$^{-2}$); and the 2021 average was $1.084 \times 10^{15}$ (molecule $\times$ cm$^{-2}$), with the annual maxima being 2.64, 2.63 and 2.65 times the minima, respectively. As shown in Figure 10, the concentration of NO$_2$ (blue curve) obviously increased with the number of fires (red curve), and like CO, the concentration of NO$_2$ also exhibited a U-shaped trend, which was more in line with the number of fires than that of CO. Figure 11 shows the correlation between the number of fire points per month and the NO$_2$ distance from the mean. The Pearson correlation coefficient between the monthly number of fire points and the NO$_2$ distance from the mean was 0.95, and NO$_2$ concentration had a stronger correlation with fire points than CO concentration because: The economy of Southeast Asia is less developed, anthropogenic activities cause less NO$_2$ emission, and most of the emissions are caused by fires. Further discussion is needed on other reasons for the stronger correlation between fire and NO$_2$ emissions.

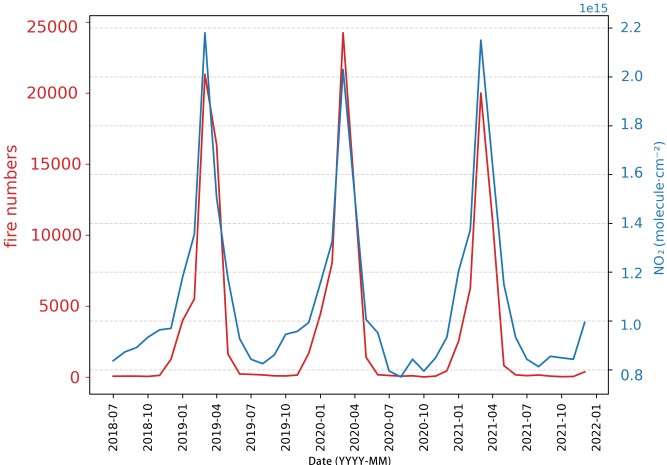

**Figure 10.** Plot of the number of fire points and NO$_2$ concentration over time.

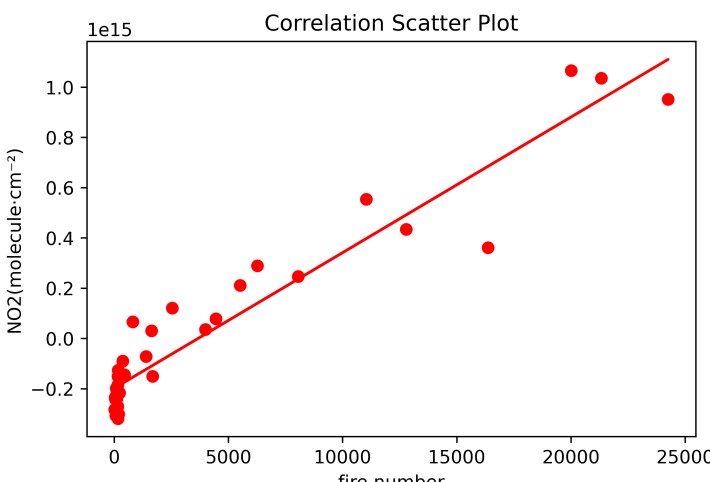

**Figure 11.** Correlation between the number of fires per month and the $NO_2$ distance level.

### 3.1.4. Changes in AOD during Fire

　　Previous studies have shown that AOD was elevated at the Kunming site in China in April 2013 due to biomass burning, and that Southeast Asia (mainly northern India, northern India–Myanmar and Bhutan) dominated the aerosol sources to the southwest of China [32]. In this paper, MODIS AOD product was selected to analyse the monthly values of AOD in the 470 nm band in PSEA during the period of 2015–2021. During the fire period, there was a small increase in AOD starting from December, and then there was an anomalous increase during the fire season (February–May), which was increased compared to the normal level by 93.5%, 228%, 228%, 247% and 135%, respectively, when compared to the normal level, with AOD reaching monthly maximum values of 0.743, 0.768, 0.535, 0.631, 0.672, 0.743 and 0.764 in March each year, reflecting the sharp drop in air quality levels during the fire season over the past seven years, as shown in Figure 12. It was not until the end of the fire season in June each year that the AOD values in PSEA returned to the normal level. However, the summer is affected by high temperature and drought, and the AOD reaches a peak again in July and August every year.

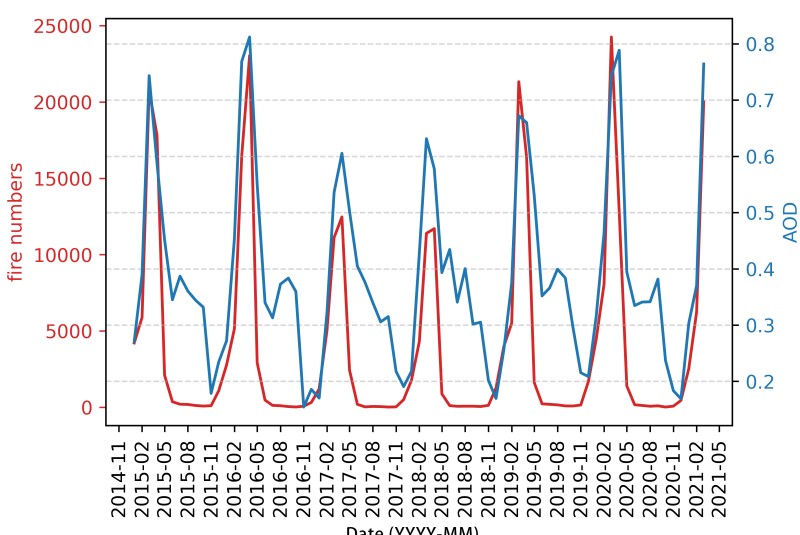

**Figure 12.** Plot of number of fires and AOD over time.

*3.2. Analysis of Numerical Simulation Results*

3.2.1. Simulation Verification

In order to better quantitatively characterize the ability of the model to simulate the atmospheric environment, we utilized four metrics, namely mean bias (MB), correlation coefficient (R), mean absolute error (MAE) and root-mean-square error (RMSE), to statistically assess the degree of agreement of the simulated results with the actual observations. MB represents the average deviation of each simulated value from the observed value, R reflects the correlation between the simulated and the observed values, MAE reflects the average of the absolute error between the simulated value and the observed value, and RMSE is used to measure the degree of deviation between the simulated value and the observed value; MB and RMSE are quantitative quantities, and the closer they are to 0, the better the simulation effect is, and the closer the absolute value of R is to 1, the stronger is the linear relationship between the two variables.

In this study, the simulation period was from 1 March to 30 March, and surface meteorological observation stations in six cities, namely Bangkok, Chiang Mai, Vientiane, Phnom Penh, Ho Chi Minh and Mandalay, were selected to validate the elements of meteorological fields, including wind speed, relative humidity, barometric pressure and the temperature at the surface 2M during the research period, and to calculate the RMSE, R, MB and MAE indicators, respectively. Table 3 lists the simulated mean values of the assessed values of each indicator during the fire pollution period. The values were also compared to reflect the simulation of the region as a whole.

**Table 3.** Model simulation of meteorological elements and validation of ground observation data.

| Elements | MB | R | MAE | RMSE |
|---|---|---|---|---|
| RH | −0.24 | 0.57 | 9.11 | 10.31 |
| T | 0.03 | 0.48 | 5.76 | 5.9 |
| P | 0.001 | 0.92 | 222.5 | 236.56 |
| WS | 0.21 | 0.31 | 0.776 | 1.007 |

The temperatures of some cities were underestimated, among which Ho Chi Minh exhibited the best result, with a simulated mean of 26.16 °C and an observed mean of 28.02 °C, with an MB of 0.00039 and an RMSE of 0.5234. Humidity plays an important role in the nonhomogeneous chemistry of particulate matter, and the humidity obtained from the simulation in the cities matched well with the observed values. Among these cities, Phnom Penh demonstrated the most accurate simulation. The simulated mean humidity in Phnom Penh was 55% and the observed mean was 59%. In terms of air pressure, the simulated mean values for each city ranged from 97,446 Pa to 101,168 Pa, MB from 0.0001 to 0.005 and R from 0.83 to 0.95, and the observed mean air pressure for each city was 101,168 Pa and the simulated mean air pressure was 100,333 Pa. Unlike relative humidity and air temperature, wind speed does not vary strongly from station to station. In terms of numerical size, the simulation results are low.

The results showed that the two trends were relatively consistent, and the fluctuations during the simulated time period were more in line with each other, with significant variations at the peaks of each elemental indicator. The maximum simulated relative humidity was too high, with a certain degree of overestimation in the actual temperature compared to the simulated temperature, which was attributed to the inaccurately simulated local meteorological fields caused by radiation, the underlying surface, and the coarse resolution of the inner grid at 27 km. The agreement was greater at the troughs of the fluctuations than at the peaks, and the modelled values were somewhat underestimated compared with the observed values.

### 3.2.2. Comparison of Numerical Modelling Results and Observed Data

Since the Sentinel-5P satellite has not yet been launched in 2016, we assessed CO concentrations during the WRF-Chem simulated fires based on MOPITT satellite measurements. The monthly average CO concentrations in the March fire season and the September non-fire season observed by the MOPITT satellite were $2.79 \times 10^{18}$ (molecule $\times$ cm$^{-2}$) and $1.14 \times 10^{18}$ (molecule $\times$ cm$^{-2}$), respectively, shown in Figure 13, meaning that CO concentration caused by fires increased by about $1.65 \times 10^{18}$ (molecule $\times$ cm$^{-2}$). However, the average CO concentration, accounting for fire emission simulated by WRF-Chem, was $2.64 \times 10^{18}$ (molecule $\times$ cm$^{-2}$); the CO concentration without fire emission was $1.91 \times 10^{18}$ (molecule $\times$ cm$^{-2}$), meaning that CO concentration caused by fires was increased by $0.73 \times 10^{18}$ (molecule $\times$ cm$^{-2}$). It suggests that the model successfully simulated the CO concentrations, and the FINN fire emission inventory slightly underestimates CO emissions, while the MEGAN biomass respiration inventory may have overestimated CO emissions, and it is recommended that these two inventories be reassessed. Comparison of the MOPITT satellite remote sensing data in the fire and non-fire seasons demonstrated that this model and inventory had a better numerical simulation capability for CO.

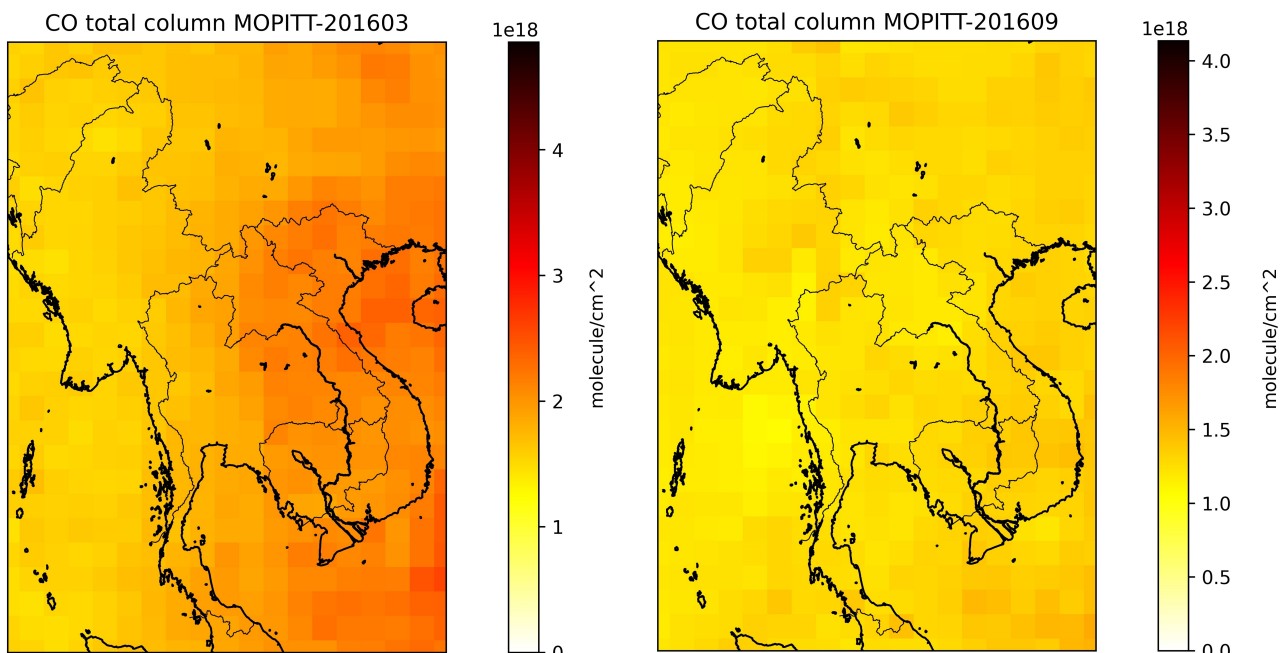

**Figure 13.** MOPITT satellite observations of monthly mean CO concentrations in the March fire season and September non-fire season.

Figure 14 shows the data from WRF-Chem with and without fires. The resolution of the satellite observations was not high, but it can be seen from the figure that CO concentrations in Myanmar and Vietnam were high, and the spatial correlation between the modelled distribution of CO column concentrations and the frequency of fires was much stronger. In addition, the area with a high CO concentration featured intense fires.

As can be clearly seen from the spatial distribution, CO concentration mirrors the intensity of fires. The spatial resolution of satellite sensors is usually at a resolution of a few hundred kilometers, which creates large uncertainty in the area of small fires, and the model reflects the fires in the area with higher accuracy. As a gas that has always reflected the carbon emissions of fire well, CO can be used to further study the fine carbon emissions of fire and the fire emissions of different pollutants by combining the satellite inversion of CO column concentration, chemical transport models, atmospheric inversion methods and the emission ratio of CO to $CO_2$ in the future [33].

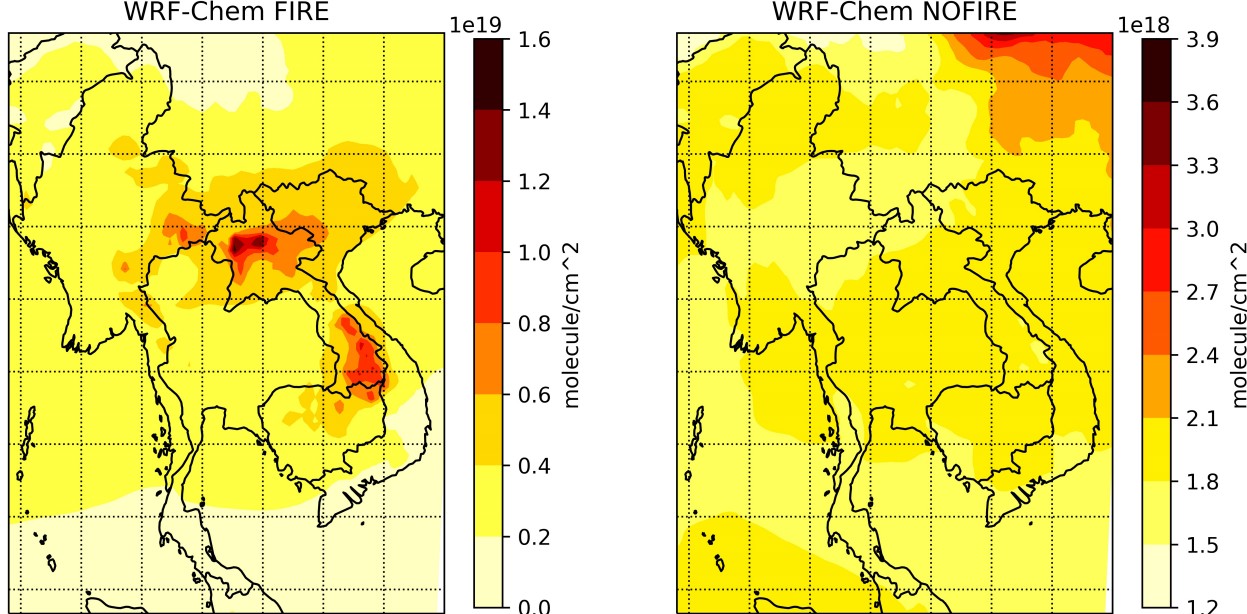

**Figure 14.** Monthly average CO concentrations for WRF-Chem with and without fire operation.

Next, we analysed the impact of $NO_2$ with WRF-Chem. The results of fire and non-fire months measured by Sentinel-5P were then used as prior conditions to assess the concentration of $NO_2$ in the WRF-Chem simulation with and without fire conditions, and the monthly mean concentrations of $NO_2$ in the March 2019 fire season and the September non-fire season observed by Sentinel-5P, shown in Figure 15, were $2.18 \times 10^{15}$ (molecule $\times$ cm$^{-2}$) and $0.86 \times 10^{15}$ (molecule $\times$ cm$^{-2}$), i.e., the mean $NO_2$ concentration in March 2019 was 2.47 times higher than that in September. The monthly mean concentrations of $NO_2$ in the fire season in March 2021 and the non-fire season in September 2020 were $1.98 \times 10^{15}$ (molecule $\times$ cm$^{-2}$) and $0.84 \times 10^{15}$ (molecule $\times$ cm$^{-2}$). The monthly mean concentrations of $NO_2$ in the fire season in March 2021 and the non-fire season in September 2021 were $2.15 \times 10^{15}$ (molecule $\times$ cm$^{-2}$) and $0.86 \times 10^{15}$ (molecule $\times$ cm$^{-2}$), respectively, and overall, the fire season $NO_2$ concentrations were 2.44 times the concentrations of the non-fire season $NO_2$ concentrations.

During the WRF-Chem simulation, the average $NO_2$ concentration including fire emissions was $1.8 \times 10^{16}$ (molecule $\times$ cm$^{-2}$), and the average $NO_2$ concentration excluding fire emissions was $4.06 \times 10^{15}$ (molecule $\times$ cm$^{-2}$), as shown in Figure 16, which shows the difference in concentration of $NO_2$ from WRF-Chem with and without fire emissions, and the difference in concentration of $NO_2$ from WRF-Chem with and without fire conditions, is about 5 times. The high concentration of modelled $NO_2$ may be due to the overestimation of $NO_2$ emissions in the FINN fire emission inventory, or it may be due to the reaction of modelled NO with $O_3$. It has been shown that NOx emitted from fires is predominantly NO, accounting for more than 90% of the emissions, and that a small portion of the airborne $NO_2$ comes from direct emissions from the source, and most of the $NO_2$ comes from the reaction of other nitrogen oxides.

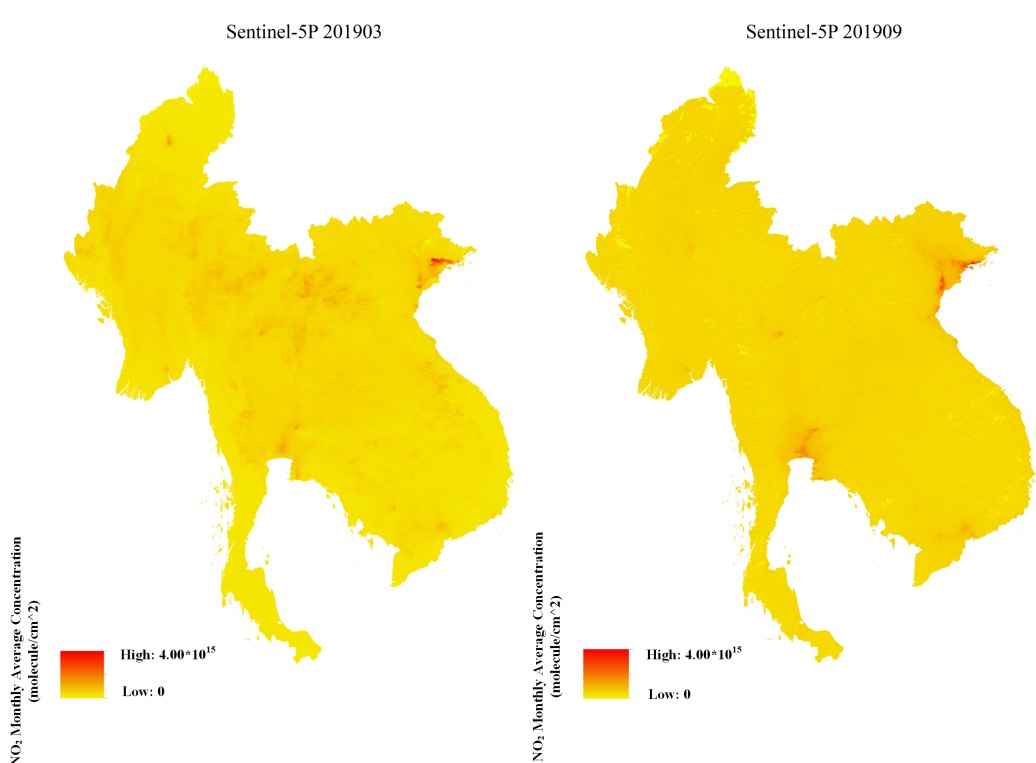

**Figure 15.** Sentinel-5P satellite-observed monthly mean concentrations of NO$_2$ during the fire season in March 201903 versus the non-fire season in September 201909.

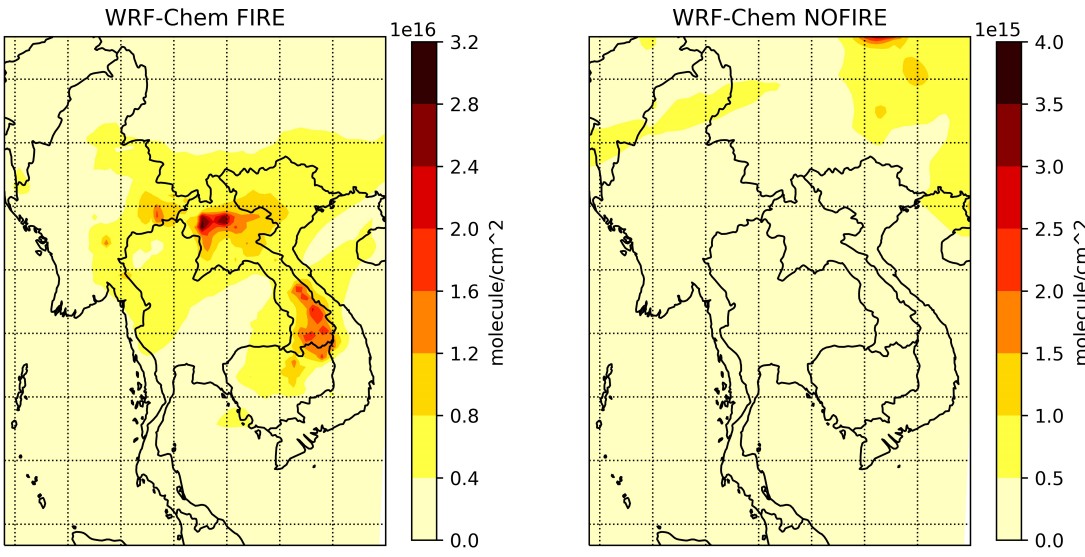

**Figure 16.** Monthly average $NO_2$ concentrations for WRF-Chem with and without fire operation.

### 3.2.3. Sensitivity Analyses: Fire Impacts on Air Quality

We compared the average monthly emissions of BC, OC, $PM_{2.5}$ and $PM_{10}$ without and with fire in the research area, as shown in Table 4. It is clear that fires release significant amounts of BC, OC, $PM_{10}$ and $PM_{2.5}$ compared to biological emissions. The concentrations of main air pollutants are reduced in both modelled scenarios, with particulate matter showing the best reduction. Advocating local residents to improve slash-and-burn farming practices and reduce biomass burning can effectively reduce particulate matter and air pollution in the local as well as the southwestern region of China.

**Table 4.** Monthly average concentrations of $PM_{2.5}$, $PM_{10}$, BC, OC.

| Conditions | $PM_{2.5}$ (µg/m³) | $PM_{10}$ (µg/m³) | BC (µg-dryair) | OC (µg-dryair) |
|---|---|---|---|---|
| WRF-ChemFire | 20.71 | 26.39 | 0.124 | 1.23 |
| WRF-ChemNoFire | 0.127 | 0.138 | $1.11 \times 10^{-13}$ | $1.81 \times 10^{-13}$ |

For the convenience of presentation, the simulation results were divided into daily units, as shown in Figure 17. In terms of daily units, the concentration of particulate matter began to rise from the seventh day, and reached a peak three times as the number of fire points increased afterwards. In the first half of March, the total number of fires per day in the study area remained below 700; the pollution of CO and $NO_2$ was in the maintenance period; and the accumulation of fires for several days reached the first peak of the concentration of CO and $NO_2$ in the middle of the simulation period. The number of fires increased in late March, with more than 1000 fires on several days, and the CO concentration reached the maximum value of 0.4 ppm during the simulation period, while the $NO_2$ concentration reached the maximum value of 0.005 ppm at the same time. The simulated trends of CO and $NO_2$ concentrations and satellite fire spot observation results exhibited good consistency and distinct variation characteristics.

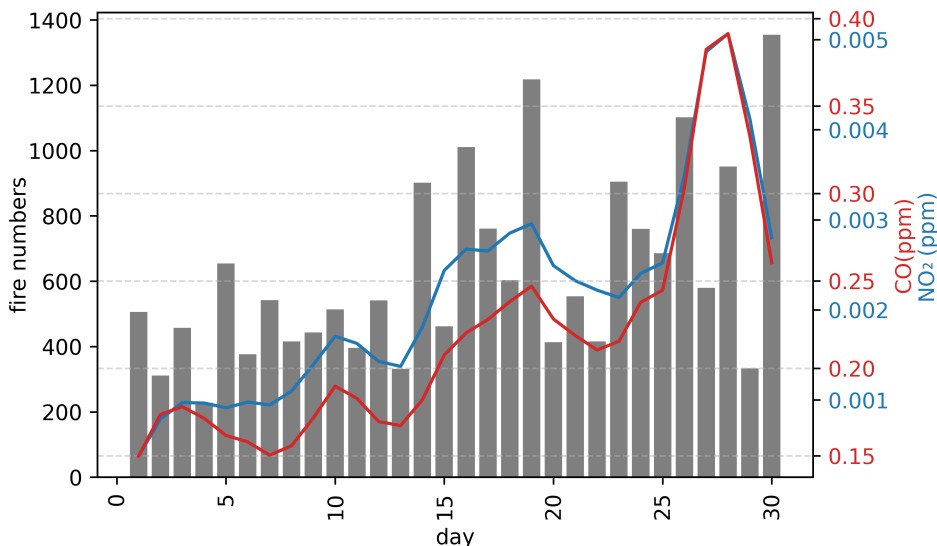

**Figure 17.** Line graph of WRF-Chem simulation results over time.

## 4. Conclusions

Based on satellite remote sensing data such as MODIS, OCO-2 and Sentinel-5P and combined the with WRF-Chem model, this work explored the trend of the concentration of atmospheric $CO_2$, CO concentration, AOD concentration, $NO_2$ concentration and spatial distribution characteristics of the five countries in PSEA by means of comparative experiments and correlation analyses. The following conclusions are obtained:

1. Satellite monitoring information shows that there are a large number of forest fires and straw burning in Southeast Asia every spring, which has an impact on the air quality in the region.There were 52,984 fires in 2015, 51,540 fires in 2016, 33,189 fires in 2017, 31,857 fires in 2018, 51,548 fires in 2019, 51,975 fire points in 2020 and 41,638 fire points in 2021.

2. The $CO_2$ column concentration in spring and summer is higher than that in autumn and winter; the $CO_2$ column concentration in autumn and winter is higher than that in summer; the $CO_2$ column concentration in autumn and summer is slightly later than the number of fires to reach the maximum value; and the $CO_2$ column concentration has a relatively significant relationship with the number of fires. The correlation

coefficient between the concentration of CO and the number of fires is 0.87, and that of the concentration of $NO_2$ is 0.95. The AOD also reflects the relationship between fire spots and air quality;

3. A control group experiment was set up to test the sensitivity of fires to CO. Satellite measurements showe a CO column concentration of $2.79 \times 10^{18}$ (molecule $\times$ cm$^{-2}$) in the presence of fires, and the model simulates a slightly underestimated CO column concentration of $2.64 \times 10^{18}$ (molecule $\times$ cm$^{-2}$) in the presence of fires, including fire emission inventories. Satellite measurements showed a CO column concentration of $1.14 \times 10^{18}$ (molecule $\times$ cm$^{-2}$) in the absence of fires and $0.73 \times 10^{18}$ (molecule $\times$ cm$^{-2}$) in the model simulation including fire emission inventories, suggesting that the MEGAN inventory needs to be assessed again. Overall, WRF-Chem is able to better simulate CO. However, the simulation of $NO_2$ is not very good.

4. The areas with high concentrations of air pollutants due to biomass combustion emissions are concentrated in the fire-prone areas (southern Myanmar and northern Laos), and their locations coincide with the distribution of fire sites monitored by satellites, as well as with the distribution of high pollutant values in the results of the model simulations.

5. WRF-Chem simulates atmospheric pollution in March. The results show that in a sustained period of increase, the concentrations of various air pollutants increase with the number of fire points. Fire pollution in this area is widespread, long-lasting and influential. It is recommended that local residents improve "slash-and-burn" farming practices and reduce biomass burning to reduce pollution and sequester carbon.

In this work, sensitivity tests were performed on biomass burning under ideal conditions, with satellite data as a control. Different factors, such as different emission categories and parameterization schemes for chemical/physical schemes, such as the precision of the selected satellite data, could contribute to the system uncertainties in this work. Owing to the scarcity of observations, particularly in PSEA nations, it is imperative to employ diverse data sets to achieve a comprehensive understanding. These nations will benefit immensely from an understanding of the effects of these biomass burns in managing and assessing their control measures. In addition to the systematic errors identified in this study, there were also large uncertainties in some other key aspects, which may also lead to differences in the models, such as the uncertainty of the jet height of the biomass combustion plume, the uncertainty of the emission factors of different species and different fires, etc. Based on the above findings, we would recommend refining the work of air quality measurement and emission inventory in the region, as it has a significant impact on the performance of the model.

**Author Contributions:** Writing—original draft, J.G.; Supervision, C.X.; Project administration, A.L. All authors have read and agreed to the published version of the manuscript.

**Funding:** This work was supported by Youth Fund of Jiangsu Basic Research Program (Natural Science Foundation) (SBK20190779), and NSFC Youth Science Foundation (42001273).

**Data Availability Statement:** Data are available upon request to the corresponding author.

**Acknowledgments:** The numerical calculations in this paper were supported and assisted by the computational support from the High Performance Computing Centre of Nanjing University of Information Engineering. This work was supported by Youth Fund of Jiangsu Basic Research Program (Natural Science Foundation) (SBK20190779), and NSFC Youth Science Foundation (42001273).

**Conflicts of Interest:** The authors declare no conflict of interest.

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
