# Peer review of "Multi-Source Satellite and WRF-Chem Analyses of Atmospheric Pollution from Fires in Peninsular Southeast Asia"

_remotesensing, doi:10.3390/rs15235463_

Round 1
Reviewer 1 Report
Comments and Suggestions for Authors
This paper is based on satellite remote sensing data and the WRF-Chem model to explore the trends and spatial distribution of CO2, CO, AOD, and NO2 concentrations in five countries of the Peninsular Southeast Asia (PSEA) of Southeast Asia. The study presents important information regarding atmospheric pollution, fires, and air quality. However, the author's English is poorly written and there are very many spelling mistakes, grammatical and semantic ambiguities, such as what is US in line 46, what is TXT, SHP, KML, and WMS in line 79, what is STL in line 180, and what is "resulsshowed" misspelled in line 230, and so on. The introduction section of this paper has poor logic. There are many articles on remote sensing observations and modeling to analyze the impacts of biomass burning on air quality in the PSEA[1-5], but the authors' introduction section clearly lacks a background survey of the current state of the research, e.g., Nuryanto and Amit Sharma's research are not relevant to the topic of this paper. Furthermore, the author uses CO observations with a 1°x1° resolution to evaluate WRF-Chem output with a 27 km resolution, which may introduce uncertainties, and this is not discussed in the paper. It's recommended that the author includes a discussion section to address research limitations and uncertainties. The paper requires significant revisions before it can be accepted in the remote sensing journal.
1.What is the reason for selecting the 2016 simulation time period for this paper? The TROPOMI provides high spatial resolution for CO and NO2, so why not select data from years with high resolution observations for simulation and comparison? Instead, the poorer spatial resolution CO (MOPITT) was selected for comparative analysis? Given the results in Figure 1, shouldn't the study focus on the year 2020? 2. Following up on the previous question, the CO resolution provided by MOPITT is too coarse compared to the spatial resolution of the model and there may be uncertainties when comparing with the simulation results, how can this be resolved? 3. Since the authors' results found strong correlations between biomass burning in the PSEA for AOD (0.76), CO (0.87), and NO2 (0.95), why did the authors analyze only the simulated CO versus MOPITT and not the simulated NO2 and AOD versus remote sensing observations? 4.Wiedinmyer, et al. [6]. found that the FINN 1.5 emission inventory might underestimate CO emissions in the PSEA in March 2016. This could introduce uncertainties into the simulation results. How did the authors consider this issue? 5. Was the smoke plume rise parameterization scheme enabled in WRF-Chem (turn on)? 6. What anthropogenic emission inventory was used in WRF-Chem? Dong and Fu [1] found that uncertainties in anthropogenic emission inventories could affect the simulation results of biomass burning in the PSEA. How did the authors address this?
7. The description of the region in the article is confusing, the title of the article uses South-East Asia, while the article uses Southeast Asia and Central South Peninsula. the author's research is only on the Peninsular Southeast Asia, while there are already a lot of scholars who use the “Peninsular Southeast Asia” to represent the region[1,7], so it is suggested that the author revise the title and unify the region in the article.
8. The analysis of Figure 2 appears before Figure 1 in the paper, which is confusing. The paper should explain the pictures in order.
Specific comments
1. Lines 33 to 49: The coverage of articles investigating the impact of biomass burning on regional air quality in the PSEA appears to be insufficient. It is recommended to rewrite this section.
2. Line 79: "TXT, SHP, KML, and WMS”?
3. Section 2.1.2: What is the data availability for all the remote sensing data mentioned in this section? How do previous studies compare these data with observations at monitoring stations?
4. Line 115: What algorithm was used to process the NO2 data obtained from the GEE platform? How was the data integrated to obtain data for the PSEA? What is the level of uncertainty associated with this Level 3 product data, and why wasn't Level 2 data used instead?
5. Line 121: The expression "Southeast Asian countries and part of our country" is not clear. Should it refer to the southern part of China?
6. Lines 157-160: The description of the dry and wet seasons in the PSEA needs clarification. Please provide additional information.
7. Line 180: "STL"?
8. Lines 223-227: The content in this section should be placed in the data introduction (2. Materials and Methods).
9. Line 242: “During the autumn and winter seasons when the weather is cold, heating is necessary for humans”. Please provide supplementary materials for this point.
10. Section 3.2.1: The section should simplify the verification of meteorological data and include validation of simulated pollutants such as PM2.5 and PM10. It would be more reliable if observed CO site data and WRF-Chem were available for the study.
11. Line 418: This section appears to be more of a discussion rather than a conclusion.
12. Figures 1 and 2 could be combined.
13. Figure 3 should include the locations of observation stations.
14. Figure 5: The CO2 label should include a subscript. What do the red and blue lines represent?
15. Figure 7 is missing a legend. What do the red and blue lines represent?
16. Figures 7 and 8 could be merged into a single figure, divided into (a) and (b).
17. Figures 9 and 10 have the same issues with the NO2 label; it should include a subscript.
18. Figure 11 is missing a legend.
19. Figures 12: The two graphs have different colorbar scales.
20. Figures 13: The two graphs have different colorbar scales.
21. Figure 14 should include national boundaries, and the colorbar scale selection appears inappropriate.
22. Figure 16: The NO2 label should include a subscript.
23. Table 1 should include information about the selection of several parameter schemes in the “namelist.input” file for the chemical module related to biomass burning simulation (turn on smoke rise plume simulation? ).
24. Table 3 should include an evaluation of wind field simulations.
25. Table 4 should include validation of PM2.5 and PM10 simulation results against observation stations.
1. Dong, X.; Fu, J.S. Understanding interannual variations of biomass burning from Peninsular Southeast Asia, part I: Model evaluation and analysis of systematic bias. Atmospheric Environment 2015, 116, 293-307, doi:https://doi.org/10.1016/j.atmosenv.2015.06.026.
2. Xing, L.; Bei, N.; Guo, J.; Wang, Q.; Liu, S.; Han, Y.; Pongpiachan, S.; Li, G. Impacts of Biomass Burning in Peninsular Southeast Asia on PM2.5 Concentration and Ozone Formation in Southern China During Springtime—A Case Study. Journal of Geophysical Research: Atmospheres 2021, 126, e2021JD034908, doi:https://doi.org/10.1029/2021JD034908.
3. Jian, Y.; Fu, T.M. Injection heights of springtime biomass-burning plumes over peninsular Southeast Asia and their impacts on long-range pollutant transport. Atmos. Chem. Phys. 2014, 14, 3977-3989, doi:10.5194/acp-14-3977-2014.
4. Zhu, J.; Xia, X.; Wang, J.; Zhang, J.; Wiedinmyer, C.; Fisher, J.A.; Keller, C.A. Impact of Southeast Asian smoke on aerosol properties in Southwest China: First comparison of model simulations with satellite and ground observations. 2017, 122, 3904-3919, doi:https://doi.org/10.1002/2016JD025793.
5. Dong, X.; Fu, J.S. Understanding interannual variations of biomass burning from Peninsular Southeast Asia, part II: Variability and different influences in lower and higher atmosphere levels. Atmospheric Environment 2015, 115, 9-18, doi:https://doi.org/10.1016/j.atmosenv.2015.05.052.
6. Wiedinmyer, C.; Kimura, Y.; McDonald-Buller, E.C.; Emmons, L.K.; Buchholz, R.R.; Tang, W.; Seto, K.; Joseph, M.B.; Barsanti, K.C.; Carlton, A.G.; et al. The Fire Inventory from NCAR version 2.5: an updated global fire emissions model for climate and chemistry applications. EGUsphere 2023, 2023, 1-45, doi:10.5194/egusphere-2023-124.
7. Ooi, M.C.G.; Chuang, M.T.; Fu, J.S.; Kong, S.S.; Huang, W.S.; Wang, S.H.; Pimonsree, S.; Chan, A.; Pani, S.K.; Lin, N.H. Improving prediction of trans-boundary biomass burning plume dispersion: from northern peninsular Southeast Asia to downwind western North Pacific Ocean. Atmos. Chem. Phys. 2021, 21, 12521-12541, doi:10.5194/acp-21-12521-2021.
Comments on the Quality of English Language
No
Author Response
Thank you very much for reading and commenting on my manuscript. They are all very professional. First of all, I will provide some explanations for the suggestion section.
- The reason for choosing the 2016 simulation time period in this article is that the 2016 Southeast Asian Haze was a transitional Haze crisis which is a recurring problem with transient air pollution bright on by fires. The fire of that year did indeed have a negative impact on the environment, affecting people's health and the economy.(https://www.bbc.com/news/world-asia-37192800)
- Since the Sentinel-5P satellite has not yet been launched in 2016, we assessed CO concentrations during the WRF-Chem simulated fires based on MOPITT satellite measurements.
- I have analyzed the simulated CO and MOPITT, as well as the simulated NO2 and TROPOMI. We generally do not compare AOD because there are errors in pattern calculation, and the AOD data is calculated from variables in the model, so the general effect is not very good.
- I agree with this comment. The monthly average CO concentrations in the March fire season observed by the MOPITT satellite were 2.79×1018(molecule×cm−2 ) , the mean value of CO concentration including fire emission simulated by WRF-Chem was 2.64×1018 (molecule×cm−2 ). The FINN emission inventory is indeed underestimated.
- The smoke plume rise parameterization scheme was enabled in WRF-Chem.
- Anthropogenic emission inventory was not used in WRF-Chem. This is a sensitivity test on fires, which does not consider the impact of human activities on the air quality changes in the area under fire and non fire conditions. This is a sensitivity test on fires, which does not consider the impact of human activities on the air quality changes in the area under fire and non fire conditions.
- I agree with your suggestion.Therefore, I have changed them.
I agree with your suggestion.Therefore, I have switched their positions.

Reviewer 2 Report
Comments and Suggestions for Authors
The quality of the English language should be improved, especially in the introduction.
Author Response
Thank you very much for taking the time to review this manuscript. They are all very professional. I have made significant revisions to the introduction and materials sections, and have also made some modifications to the sentences in the third section.

Round 2
Reviewer 2 Report
Comments and Suggestions for Authors
The English Language has been improved.
Author Response
|
Thank you very much for taking the time to review this manuscript. Please find the detailed responses below and the corresponding revisions/corrections highlighted/in track changes in the re-submitted files. |
